# Comparative Computational Modeling of the Bat and Human Immune Response to Viral Infection with the Comparative Biology Immune Agent Based Model

**DOI:** 10.3390/v13081620

**Published:** 2021-08-16

**Authors:** Chase Cockrell, Gary An

**Affiliations:** Department of Surgery, Larner College of Medicine, University of Vermont, Burlington, VT 05405, USA; Robert.cockrell@med.uvm.edu

**Keywords:** comparative biology, computational biology, mathematical modeling, innate immunity, bats, viral tolerance, agent based model, inflammasome, zoonotic transfer, viral pandemic, COVID-19

## Abstract

Given the impact of pandemics due to viruses of bat origin, there is increasing interest in comparative investigation into the differences between bat and human immune responses. The practice of comparative biology can be enhanced by computational methods used for dynamic knowledge representation to visualize and interrogate the putative differences between the two systems. We present an agent based model that encompasses and bridges differences between bat and human responses to viral infection: the comparative biology immune agent based model, or CBIABM. The CBIABM examines differences in innate immune mechanisms between bats and humans, specifically regarding inflammasome activity and type 1 interferon dynamics, in terms of tolerance to viral infection. Simulation experiments with the CBIABM demonstrate the efficacy of bat-related features in conferring viral tolerance and also suggest a crucial role for endothelial inflammasome activity as a mechanism for bat systemic viral tolerance and affecting the severity of disease in human viral infections. We hope that this initial study will inspire additional comparative modeling projects to link, compare, and contrast immunological functions shared across different species, and in so doing, provide insight and aid in preparation for future viral pandemics of zoonotic origin.

## 1. Introduction

The potential for zoonotic transfer of viruses from bat populations to humans is highly significant in terms of viral pandemics. Bats have been implicated as zoonotic reservoirs/precursors for Ebola [1], Middle East respiratory syndrome coronavirus (MERS-CoV) [2], severe acute respiratory syndrome coronavirus 1 (SARS-CoV1) [2] and, most recently, severe acute respiratory syndrome coronavirus 2 (SARS-CoV2) [3]. The public health impact of these pandemics has prompted a great deal of interest in examining specific aspects of bat physiology and immunology to gain insights into why bat-borne viruses might be particularly pathogenic when they adapt to cause human disease [2,3,4,5]. One of the unique aspects of bats among mammals is their ability to engage in active flight, and therefore there has been an interest in examining the evolutionary adaptations that have arisen in bats to accommodate the increased metabolic demands associated with active flight, particularly with respect to the configuration of their immune systems [3,4,5]. Specifically, the increased metabolic demands associated with flight suggest the need to adapt to increases in oxidative stress and associated cellular damage [3,4,5], but at the potential cost of impairing their response to certain types of microbial infections. This, in turn, would suggest adaptations of other means of controlling infectious insults. The combination of these two seemingly required features, (1) increased resistance to cellular damage and oxidative stress and (2) parallel augmented anti-infection capabilities, has led to targeted investigation and characterization into two biological modules that address these features: (1) differences in bat inflammasome activation, and (2) constitutive production of type 1 interferons, a notable antiviral mediator.

The inflammasome is an intracellular multi-protein complex that processes danger signals (either microbial products or evidence of cellular damage) into a series of inflammatory response processes, including the production and release of proinflammatory cytokines (notably interleukin-1 (IL-1) and interleukin-18 (IL-18)), the induction of nuclear factor kappa-b (NF-κB), a transcription factor responsible for inducing a series of inflammation-related mediators, and promoting pyroptosis, a form of programmed cellular death that is itself an inflammation-promoting process [6,7,8]. As such, the inflammasome is a critical component of an organism’s response to effectively deal with infection or tissue damage but, if dysregulated, can contribute to the pathophysiological manifestation of a host of diseases [6,7,8]. Specifically, aberrant inflammasome activation has been invoked as a major source of disease severity in viruses of bat origin, where the “cytokine storm” resulting from extensive systemic inflammation produces collateral tissue damage and organ dysfunction [6]. Alternatively, bats have been demonstrated to have significantly suppressed inflammasome activation and activity, perhaps because of their need to deal with periodic episodes of high metabolic stress that can produce the exact mediators that would lead to priming and activation of the inflammasome. These adaptations include resistance to inflammasome priming from extracellular danger signals, such as reactive oxygen species, pathogen/damage-associated molecular patterns (P/DAMPs) and proinflammatory cytokines, and decreased transduction of subsequent inflammasome activation leading to the release of IL-1/IL-18 and induction of pyroptosis [3,4].

In concert with decreased inflammasome activity, bats have enhanced antiviral capabilities in terms of their production of type 1 interferons (T1IFNs). T1IFNs are associated with intracellular compounds that inhibit viral replication and assembly, as well as serving an extracellular role in terms of recruiting immune cells, notably natural killer cells (NK cells) and antigen-presenting cells (macrophages and dendritic cells) to eliminate infected cells. In contrast to most mammals, where the production of T1IFNs is primarily initiated by evidence of infection, some bat species constitutively produce T1IFNs [3,4]. This provides these bats with an enhanced baseline level of antiviral capability, which, in conjunction with their dampening of inflammatory reactivity, allows them to have improved tolerance to viral insults.

While there are other mechanisms for viral resistance in bats, the presence of features of disordered inflammation manifesting in human disease represents a natural focus on the difference in inflammasome activity between humans and bats, and such investigations should take into account the antiviral compensatory mechanisms of differential T1IFN behavior between humans and bats. The work represented in the above series of references demonstrates beneficial insights from the traditional approach to comparative biology, but we believe that this comparative process can be enhanced by dynamically representing the putative hypothesized mechanisms such that the consequences of those mechanisms can be seen. We have termed this use of computational modeling as *dynamic knowledge representation* [9,10] as a useful label to describe the difference between a dynamic computational model and a static picture/diagram. This process, which is common to virtually all types of dynamic computational/mathematical modeling, takes as its starting point the standard diagram ubiquitous in biomedical literature that depicts a series of biological entities connected by arrows that suggest their causal relationships. While useful for gaining an overview of the components and their interactions for a particular biological system, these diagrams are static representations of knowledge. Dynamic knowledge representation via mechanism-based mathematical/computational modeling takes those diagrams and “brings them to life.” The components and interactions present in the diagram are converted to equations/computer code, and as the model executes those, the dynamics of those interactions are visualized via simulations. There are several benefits of this use of computational modeling:It can be difficult to intuit the overall effects of a complex set of interactions, and therefore a dynamic representation of this knowledge can uncover unanticipated or paradoxical dynamic effects, particularly when positive and negative feedback are present [9].While these models may be complex, they are necessarily selective abstractions of the real world (as are the diagrams they are based on). Since the model will only exhibit behaviors possible from the interactions they embody, they can be used to determine the sufficiency of a particular hypothesis/theory in terms of explaining observations from the real world. While they cannot falsify a particular hypothesis, they can suggest where additional features of the biology need to be added to achieve a particular desired behavior.Toward this end, computational models of this type can be used for examining new hypotheses, be they components or interactions within the system itself or putative interventions [9]. Examining the plausibility of such modifications can suggest new experiments or observations that might need to be performed.These types of computational models can be used as experimental objects, where simulation experiments can be performed that evaluate the new hypotheses noted in #3. Since simulation experiments are not subject to the cost and logistical constraints that apply to wet-lab experiments and can often be run at a considerably greater scale (in terms of number of individuals examined), they can be a useful adjunct to pre-testing and design of subsequent real-world experiments [9].As computational embodiments of knowledge/hypothesis structures, these models can serve as “bridging” knowledge structures that can represent what is conserved from one individual or zoological context to another. Rather than relying on a list of components and features, a computational model for dynamic knowledge representation can encapsulate what is functionally “similar” from one species/organism/individual and note what the effect might be for what is explicitly “different.” [11].

Although all computational models employ some degree of necessary abstraction (e.g., it is not possible to incorporate every possible aspect of knowledge into a model), we assert that there are investigatory benefits for constructing and using complex computational representations of biological knowledge, if for no other reason that such models embody what biological researchers consider important enough to expend their energies investigating. The ubiquitous diagrams of biological processes express their level of detail for a reason: those details are of interest to biological researchers. We do not separate the modeling endeavor from the overall scientific process and therefore tailor the representational level of our models to what members of the research community are interested in, as reflected by the content of the diagrams they choose to use in their publications. Furthermore, given the necessary abstractions present in these types of models, there is no supposition that they represent ontological truth. They are expected to be insufficient at some level, but ideally transparently so, thereby helping guide further scientific investigations. To paraphrase the famous quote from George Box that “All models are wrong, some models are useful” [12], useful models are those that are invalid in such a way that convinces a wet-lab researcher to perform a new experiment.

It is also very common to use computational-mathematical models to link data acquired in one species to provide insights into behaviors present in others (e.g., humans). This is an extension of the standard experimental process of using model organisms (i.e., yeast, worms, mice, non-human primates) to investigate mechanistic hypotheses/knowledge in a more controlled fashion and extrapolating those findings to human biology. However, since this standard process focuses on being able to translate functions that are presumed to be conserved from one species to another, this workflow necessarily focuses on what is presumed to be similar across these systems. It is far less common to explicitly target with computational models what is mechanistically different between different species, with the motivation that there may be some insight into how those differences manifest, and thereby can guide the development of new therapies. In our review of the literature, we have not been able to find other examples of dynamic mechanism-based modeling used in this fashion, and certainly not with respect to bats and humans.

Toward this end, we present an agent based model of the innate immune response that encompasses and bridges the differences between bat and human response to viral infection; we term this model the comparative biology immune agent-based model, or CBIABM. To our knowledge, this is the first dynamic mechanism-based computational model that seeks to directly compare bat and human immune mechanisms and the consequences of those mechanistic differences, namely inflammasome activity and differences in T1IFN dynamics, in terms of manifesting disease.

## 2. Materials and Methods

### 2.1. Overview and Abstraction Level

The CBIABM is an agent based model (ABM). Agent based modeling is a discrete-event, rule-based, spatially explicit, computational modeling method that represents dynamical systems as populations of stochastically interacting, rule-based components, termed *agents* [13]. The structure of ABMs is well suited to representing multi-cellular biological systems, with cell types represented by classes of agents, where the cellular agents are governed by rules extracted from basic science literature regarding the causal mechanisms (that arise from natural laws) that govern cellular behavior. As such, agent based modeling has become extensively used in computational biology for examining a wide range of biomedical processes, such as sepsis [14,15,16], infectious disease [17], wound healing [18,19], vascular biology [20], and cancer (for an overview, see [21]). Specifically, we contend that ABMs are well suited for the task of dynamic knowledge representation, with the added benefit of modularity in terms of agent classes and hierarchical aggregation (i.e., ABMs of specific tissues can be combined to represent more complex organ structures) [9].

As noted previously, the selection of the level of abstraction is a key component of model development and should be guided by the intended purpose of the model. The CBIABM is intended to evaluate the system-level consequences of differences between humans and bats among several functional modules in the immune response to viral infection. The functional modules involved are not represented with high levels of molecular detail but rather treated as aggregated input-output processes; thus, we avoid the need to incorporate specific molecular kinetic rate constants and instead focus on being able to represent the general time-courses of the outputs of these processes. For example, we recognize that the inflammasome is a complex aggregation of various proteins, signaling molecules, and transcription factors; however, functionally, these details can be reduced to representing a set of cellular inputs that trigger priming and activation of this complex, and a series of defined cellular behaviors that result from its activation.

Similarly, for the current set of simulation experiments on the CBIABM, we do not represent a specific type of virus; the intent is to evaluate the differing immune configurations between bats and humans in a virus-agnostic fashion. Therefore, the abstracted virus has a probabilistic ability to invade a cell, a specified replication rate, and viral release modeled as an abstracted exocytosis function that will eventually cause cellular death unless the cell dies by apoptosis beforehand. These abstracted viral functions have been implemented to allow for future modification to specific virus types, but for this initial study, we model them generically. This type of structure also allows for the future implementation of more detailed viral properties, such as receptor-specific binding, apoptosis inhibition, and other immune-evading capabilities.

Also, given the goal of examining the immune differences between bats and humans, we focus on the innate aspects of the immune response in the current development of the CBIABM. This is because the high metabolic rate in bats has been identified as a main driver for what makes them immunologically different, and the ability to deal with intrinsic cellular damage and byproducts of increased oxidative/metabolic stress is primarily carried out by the innate immune response [3,4]. While we recognize the importance of adaptive immunity in human disease, particularly regarding the development of cytokine storm in severe clinical illness, in terms of comparing the role of the inflammasome and baseline antiviral capabilities we make the modeling decision that focusing on the innate response is sufficient for this initial examination of the transition from bat viral tolerance to human viral pathophysiology.

### 2.2. Model Implementation

The CBIABM is implemented in NetLogo [22], a freeware software package used for a range of agent based modeling tasks ranging from K-12 education to modeling of complex real-world systems. One of the primary benefits of NetLogo is that the models are written in a human-friendly readable coding language. For interested readers, the entire CBIABM (with internal documentation) can be downloaded as a NetLogo executable file from http://www.github.com/An-Cockrell/Comparative-Biology-Immune-ABM (accessed on 13 August 2021). This folder includes the NetLogo executable file for the CBIABM, which includes documentation of the code and, through the Behavior Space function, the parameter and experimental settings used for the data presented in this paper. The Github folder includes instructions on how to use the CBIABM in a file named “Supplementary Material Text S1-published”, and this information is also included as a supplement to this paper, “Appendix A: Instructions for running the CBIABM”. The topology of the CBIABM is a two-dimensional square grid composed of 51 × 51 “patches” (to use NetLogo terminology) where the edges of the grid wrap such that a cell moving off one edge appears on the opposite edge (right-left, top-bottom). Each patch represents a spatial container for extracellular entities (mediators, extracellular viral particles) and cellular agents that can occupy that area of space. NetLogo incorporates several core commands (“primitives” in NetLogo terminology) that are readily employed for constructing cell-level ABMs. Two of these merit specific mention here, as they relate to how rules for cellular behavior are implemented and may aid understanding in the rule descriptions to follow:“Diffuse”: This primitive simulates diffusion of a patch variable to its surrounding 8 patches. The argument for the primitive is the percentage of the value of the center patch that is removed and then evenly distributed among the surrounding 8 patches.“Uphill”: This primitive is essentially a chemotactic function, where an agent surveys the surrounding 8 patches and moves toward the patch with the highest target variable. Note that within the CBIABM, this primitive is decomposed into several lines of code to allow chemotaxis to be driven by combinations of mediators. In the CBIABM, all mobile cells share the same movement rate.

The CBIABM runs in a step-wise fashion during which all the functions of the model are executed, where each step represents approximately 10 min of real-world time; the parameters and rules for the CBIABM are thus fitted to generate behaviors in realistic time courses based on this step interval. Note that the “virus” reflected in this current version of the CBIABM is a generic virus and not calibrated to any particular viral species; however, the modular rule and parameter structure of the CBIABM is such that specific types of viruses can and will be implemented in the future. Each simulation run represents an infected simulated patient/animal. The high-level sequence of events are as follows:The simulation is initialized;A specified amount of extracellular virus is placed in a random pattern onto the epithelial cell grid;The rules for the cells are executed. The implementation of NetLogo shuffles the order of execution within each class of agent but runs each class of agents in a specified sequence. Once the specified set of instructions for each step are completed, the process starts again until the stopping conditions are met;The total time course simulated in the experiments below is 14 days (though this can be modified).

There are stochastic processes incorporated into the rules of the CBIABM: (1) they manifest in the initial distribution of cells and extracellular virus, (2) to provide some noise in terms of the rate of viral infection and replication, and (3) to provide noise for incorporated thresholds for cellular rules related to the activation status of the cells. These stochastic components result in each run being unique and distinct, mimicking the heterogeneity seen across a population of organisms, be they bat or human. Therefore, the evaluation of the output of the CBIABM is performed at the population level, with evaluation metrics reported as a distribution of a particular model output across a population of simulation runs. The primary output metric for an individual run is “%System-Health”, which represents the health of the system as the percentage of healthy epithelial cells among the total possible number of 2601 (51 grid spaces × 51 grid spaces). Note that this output metric does not distinguish between epithelial cell agents that are eliminated by apoptosis or necrosis; the supposition is that the viability of a tissue is a function of the number of healthy cells present. This view of tissue damage/health also assumes that the different tissue-level consequences of the various forms of cellular death: apoptosis, pyroptosis, and necrosis, are due to their relative ability to induce and propagate inflammation, which reduces the total number of healthy epithelial cells (as opposed to a view that cells that die via necrosis are somehow “more dead” than those that die by apoptosis). In addition, note that in the current version of the CBIABM, we elected to not represent the dynamic turnover of macrophages, NK cells, and dendritic cells, with the result that population levels of these cell types are held constant for the duration of the simulation runs. While we recognize that recruitment and depletion are present in these cell populations in the real world, we believe that the goal of the current modeling project, which is to demonstrate significant and qualitative differences between bat and human resistance to viral infection, is not substantively affected by this abstraction. The rationale for this choice of modeling abstraction are as follows:While the initial recruitment of immune response cells may enhance the initial containment of the viral inoculum, these cells accomplish that effect by the generation of proinflammatory mediators, which are the exact processes invoked in disease manifestation that bats are thought to avoid. Therefore, while the addition of this feature might alter the specific potential transition zone in the difference between bat and human disease manifestation is should not qualitatively alter the existence of such a zone, which is what these simulation experiments are intended to show.Similarly, while depletion and exhaustion of inflammatory cells are known to occur in severe human disease, these processes only occur after the inflammatory response to the viral inoculum has substantially progressed. Since the hypothesis underlying bat resistance to viral infection emphasizes early attenuation of the inflammatory response, this phenomenon does not affect those early dynamics and the sought-after qualitative difference between bat and human disease manifestationAlternatively, polymorphonuclear neutrophil (PMN) populations do dynamically change in the CBIABM. This is because there are no baseline PMNs in non-inflamed tissue, and therefore their recruitment must be explicitly represented (in contrast, macrophages, NK cells, and dendritic cells, though augmented by circulating precursors, are already present in tissue at baseline).

Therefore, our decision to hold macrophage, NK cell, and dendritic cell populations steady represents an abstraction of the trade-off between recruitment and depletion that we do not believe will affect the intended goal of this modeling project. See the text on calibration in Section 2.3 for an example of how we would have reassessed this modeling decision during development had this decision impacted our ability to achieve the desired behavior of the CBIABM.

The CBIABM represents the following cell types listed below. Their behaviors and properties are described in plain language for clarity; interested readers are encouraged to download the actual model to see how these properties have been implemented into NetLogo code.
1.Epithelial cells. These are generic epithelial cells that represent the cells initially exposed and susceptible to viral infection. The epithelial cells also abstractly represent generic “tissue”, as the current CBIABM does not explicitly represent muscle or specific organs. One epithelial cell agent occupies a single grid space for a total of 2601 possible healthy epithelial cells in the system, and they do not move. The total health of the system is reflected by the percentage of healthy epithelial cells out of 2601, the variable called “%System-Health”. The epithelial cell agents have the following variables and functions:
a.Susceptibility to infection: this variable represents how readily the epithelial cell can be infected by extracellular virus and abstracts its expression of potential receptors that can be targeted by various viruses. This is a constant in the current version of the CBIABM and is used to probabilistically (in relation to the number of extracellular viral particles on that specific gride space) determine whether an individual epithelial cell agent becomes infected.b.Susceptibility to reactive oxygen species/cytotoxic compounds: this variable represents how much damage the epithelial cell can sustain before it undergoes necrosis, a proinflammatory form of death that results in the production of additional danger signals (P/DAMPs).c.Metabolic Byproduct (Met-By): this variable represents the amount of oxidative byproducts produced from baseline metabolism and are sensed as P/DAMPs. This value is 10-fold higher in bats versus humans, representing the increased metabolism in bats needed for flight [3]. This value is added to the P/DAMPSs present on a particular patch.d.Total cell membrane: this variable represents how much cell membrane the cell has that can be consumed by viral exocytosis before the cell dies by necrosis (proinflammatory death) and release of danger signals (P/DAMPS).e.Apoptosis: This function is initiated by sensing of viral infection and represents programmed cell death to shorten the time (and therefore total amount) of viral production by an infected epithelial cell. This function is also accelerated by interactions with NK cells as a representation of NK cells’ antiviral effect. As a result of apoptosis, almost no epithelial cells progress to membrane consumption death (1c above), though may be altered in future versions where for simulations of viruses known to interrupt apoptosis. Notably, cells that die by apoptosis do not propagate inflammation; this is distinct from cells that die via necrosis (1d above), which release P/DAMPs until they are cleared by phagocytosis by macrophages.f.Production of type 1 interferons (T1IFN): primary inflammatory and antiviral mediator produced by infected epithelial cells in humans and at baseline in bats.g.Regeneration: This is an abstracted healing function that allows the regrowth of new epithelial cells into empty patches from where dead epithelial cells (either apoptotic or necrosed) have been cleared by phagocytosis. This process is simulated to take 3 days.
2.Natural killer cells (NK cells): These are mobile immune cells that are a major component of innate antiviral activity. They migrate toward infected epithelial cells and accelerate their apoptosis. Their functions and properties include:
a.Chemotaxis to T1IFN. This means they move up a gradient of T1IFN.b.Production of interferon-gamma (IFNg) in the presence of T1IFNs, interleukin-12 (IL-12) and interleukin-18 (IL-18).c.Accelerate apoptosis in infected epithelial cells via abstraction of perforin and granzyme function.d.NK cell populations are held constant for the duration of the simulation (see discussion/rationale in the preceding text).
3.Macrophages: These are mobile immune cells that respond to signals produced by infected epithelial cells and immune cells (NK cells, dendritic cells, polymorphonuclear neutrophils, and other macrophages). These cells are central regulators of the innate immune response; they also phagocytose extracellular virus and cellular debris from dead epithelial cells, clearing space for new epithelial cells to regenerate into. Their functions and variables include:
a.Chemotaxis to T1IFNs and P/DAMPS;b.Have a differential activation level representing the ability to perform proinflammatory functions (M1 phenotype) or anti-inflammatory functions (M2 phenotype). The activation level is determined by a balance between proinflammatory signals: T1IFN, P/DAMPs, IFNg, and IL-1 versus anti-inflammatory signals: Interleukin-10 (IL-10);
i.M1 Macrophages produce interleukin-8 (IL-8) and Il-12, and if their inflammasome is activated, tumor necrosis factor (TNF), interleukin-6 (IL-6), IL-10 and IL-1 and IL-18;ii.M2 macrophages produce IL-10.
c.Have an abstracted inflammasome that becomes activated through two steps:
i.Priming: Occurs when exposed to P/DAMPs or TNF;ii.Activation: If already primed, if sufficient extracellular virus is phagocytosed, this triggers the production of precursors for IL-1 and IL-18, allows M1 production of TNF, IL-6, and IL-10, and initiates the pyroptosis pathway.Both priming and activation have defined thresholds; these are set at different levels between humans (lower) and bats (higher).
d.Can undergo pyroptosis: as discussed above, pyroptosis is an inflammogenic form of cell death. The precursors to IL-1 and IL-18 are released as active cytokines upon pyroptosis, as well as the production of P/DAMPs representing the release of extracellular DNA at cell death. Pyroptosis occurs ~2 h after inflammasome activation [8,23,24,25]. Note that we have elected to keep macrophage populations steady for the duration of the simulation; therefore, when a macrophage undergoes pyroptosis, it is immediately replaced by the creation of a naïve macrophage placed randomly in the world grid. This is a modeling decision that abstracts but qualitatively reproduces steady-state depletion/recruitment dynamics.e.Perform phagocytosis: this is the endocytosis of extracellular viruses and cellular debris, clearing away damaged cells to allow for epithelial regrowth. However, it is recognized that there is a limit to the amount of material a macrophage can phagocytose; therefore, there is a variable that determines the upper limit of this amount: when this limit is reached, the macrophage is considered “exhausted” and is unable to clear any more material [26].
4.Dendritic Cells: These are antigen-presenting cells that have similar functions to macrophages but are a key component in the transition from innate to adaptive immunity: they are primarily responsible for presenting antigen to naïve T-cell subtypes and inducing antigen-specific differentiation of various T-cell subtypes. However, in the current version of the CBIABM, their role in inducing adaptive immunity is not represented, and rather they serve primarily to chemotax in response to T1IFN and produce IL-6, IL-12, and IFNg. As with macrophages and NK cells, dendritic cell populations are held stable for the duration of the simulation.5.Endothelium: The majority of the initial viral response is considered to take place in the epithelial tissue at risk, though the vascular supply to that tissue means that there is close proximity with the endothelial cells lining those vessels. The activation of the endothelium is a key step in the transition from what would otherwise be an inflammatory process restricted in local tissue to an expansion toward systemic effects. As such, endothelial activation is a critical tipping point that affects the manifestation of the disease [27]. The inflammatory role of the endothelium is represented in the CBIABM by projecting a layer of endothelial cells “on top of” each epithelial cell and having those cells being able to be activated by signals generated on the patch on which it sits. Endothelial cells are activated by a combination of inflammatory signals: IL-1 and TNF. This activation involves the function of the endothelial inflammasome, which in humans activates a series of signaling and adhesion functions that eventually lead to the recruitment of polymorphonuclear neutrophils (PMNs) from the bloodstream to the area of inflammation. We *hypothesize* that, similar to their immune cell inflammasomes, bats have a reduced degree of responsiveness in their endothelial inflammasomes; this decreased function is reflected in the different values assigned to the endothelial activation threshold between human and bat simulations. In addition to the simulated induction of adhesion molecules, activated endothelium produces platelet-activating factor (PAF), which is a chemotaxis signal for polymorphonuclear neutrophils (PMNs) [28].6.Polymorphonuclear neutrophils (PMNs): these are the primary circulating immune/white blood cells. They serve a central role in response to bacterial infections, but in general, their role in containing viral infections is less pronounced. However, they can be recruited to sites of viral infection if the endothelium in the region of the infection becomes activated and initiates the sequence of PMN adhesion and migration through the blood vessel walls into the tissue. The process of PMN adhesion and migration is abstracted in the CBIABM and is modeled to take about 6–12 h from initial endothelial activation through the production of endothelial adhesion molecules, the adhesion of PMNs to those molecules through to the migration of PMNs into the tissue. Once in the tissue, they will chemotax toward IL-8 and PAF and subsequently undergo a respiratory burst that produces proteolytic and cytotoxic compounds that induce epithelial cell necrosis (a proinflammatory cell death) [29]. Respiratory burst is associated with the death of the PMN. As noted above, PMN populations are dynamically shifting via recruitment via activated endothelium and therefore depleted by the respiratory burst process.

Mediators are generated and modified by the cellular agents based on their programmed rules and are represented as variables present on each grid space; therefore, the total system-level amount of a particular mediator is the sum of all the individual values for each patch. Mediators diffuse from patch-to-patch using the NetLogo “diffuse” function, where a defined percentage of the variable is evenly distributed to the surrounding 8 patches. Mediators are reduced by three processes:They are consumed by binding to receptors on the cellular agents.There is a percentage degradation of the value of each mediator per step.There is an arbitrary lower threshold of the value of each mediator, at which point the value is set to 0.

The mediators included in the CBIABM, their represented functions, and associate references can be seen in Appendix B Table A1. Mediator levels are updated using cellular rules that arithmetically increment or decrement their value each step per patch; the numerical values that are used to update the mediators are the “parameters” in the CBIABM. We also note that the same mediator can be manipulated by multiple cell types (as can be seen in Appendix B Table A1 and Appendix C Table A2). Details as to how this model rule structure undergoes calibration are relayed in Section 2.3 Simulation Experiments. As with the cellular rules, interested readers are encouraged to download the actual model to see their exact implementation in NetLogo code. Appendix C Table A2 is a grid that links mediators to the cells that produce them.

Initialization of the CBIABM involves generating epithelial cell agents that populate each grid space of the 51 × 51 square grid. Set populations of NK cells, dendritic cells, and macrophages are randomly distributed across the space. A viral inoculum (that can be varied) is applied by setting the value for extracellular virus to 100 +/− Random 20 (to introduce noise) for a random number of grid spaces equal to the initial inoculum variable; this is the variable that represents the perturbation to the system. During simulation runs, the cells and mediators listed above will interact with each other based on their programmed rules but acting upon different local conditions arising from the heterogeneous spatial distributions of the initial configuration of the model’s world (i.e., inflammatory cell and extracellular virus locations). Figure 1 depicts the set of interactions between the various cellular agent types, the simulated mediators (production and effects), and functions represented by the model rules. As noted previously, the entire CBIABM version that was used to generate the data for this paper is available for download; this includes descriptions of the specific parameters used for the simulation experiments described below.

### 2.3. Simulation Experiments

Given the goal of examining fundamental differences between bat and human responses to a viral infection (e.g., not resistance to specific viruses), parameters related to viral invasiveness and viral replication rate were kept constant between simulations in the bat and the human. This choice was made to remove virus-specific properties, such as exploitation of specific cellular receptors as a means of producing differential virulence, as this would confound the evaluation of more general and conserved differences between bats and humans. The time course for the viral replication was set to generate peaking or at least plateauing mediator levels between 3 and 7 days simulated time, corresponding to the onset of symptoms ~3–7 days after inoculation. We make the modeling assumption that the basic interaction map (seen in Figure 1) is conserved between bats and humans, albeit with the notable differences listed here:Bats have higher baseline production of P/DAMPS to simulate the enhanced metabolic rate needed for flight (reflected by the difference in “metabolic byproduct” value)Bats have baseline production of type 1 interferonsBats have enhanced intracellular antiviral effect of type 1 interferonsBats have reduced inflammasome priming and activation (reflected by higher values for the variables “inflammasome-priming-threshold”, “inflammasome-activation-threshold”, and “bat endothelial activation” versus “human-endothelial activation”).

Our goal is to evaluate a minimally sufficient set of differences that have been emphasized in recent reviews concerning the comparative immunology between bats and humans regarding their respective responses to viral infection [3,4]; future expansion of the CBIABM can incorporate additional differences between the species.

The initial calibration of the model focused on matching behavior to the general dynamics of human infection. The reason the human behavior was targeted at first is because it is the human response to viral infection that exhibits the full dynamic range of system outcome, i.e., from no or minimal disease severity to considerable disease severity. Since bats are noted to not manifest disease from infection, there is no range of system-level bat phenotypes for the model to reproduce if the calibration was attempted for them first. Therefore, we proceeded with calibration to general human dynamics, then held model parameters constant except for those features explicitly noted to be different in bats. As noted above, the rule structure of the CBIABM consists primarily of arithmetic increments or decrements of mediator values by some constant value (“parameter”) that occur per step. Given this model rule structure, there is not a single equation that governs the dynamics of a particular mediator; rather, the mediator’s dynamics arise from the sum of the interactions present in the entire model. As such, this is not a modeling approach that needs to use aggregated kinetic rate constant parameters (of the type primarily reported in wet-lab experimental results) as calibration targets since there are no kinetic equations present. Additionally, since the goal of this modeling project is to compare the overall dynamic characteristics of the various simulations, which depend on the relative relationships between the rules and their respective parameters, and not specific experimental values, the variable outputs of the CBIABM are unitless. We have designed the CBIABM so that future work that aims to represent specific viral infections and/or reproduce specific experimental conditions can be readily performed using our established strategy for large scale parameter space fitting/characterization to specific data sets [30], but that is not the goal of the current work. Therefore, calibration involved modification of parameters by hand-fitting to expected general time courses of viremia, viral eradication, initiation of inflammatory responses and general trajectories of cytokine time courses with the modeling assumption that symptoms arising from proinflammatory cytokines would manifest ~3–7 days post-inoculation. Hand-fitting is a standard practice that involves observing the behaviors of the CBIABM given a particular set of parameter values. Then, the modeler determines if the desired behavior is acceptable. If it is not, then the modeler determines which parameter they what to adjust to move the model toward a more desirable behavior. Since the behavior of the CBIABM is dependent on the relationship between the variables in the model, it is necessary to make several initial choices to serve as reference points for further adjustment. The three used in this case were:The world size was chosen to consist of a 51 × 51 grid. This was arbitrary and made for computational efficiency based on our experience constructing this type of model.The time frame was selected such that one step of the CBIABM represents 10 min; this provided a frame of reference for how the various cellular rules manifested their dynamics.The initial populations of macrophages, NK cells, and dendritic cells were set at 50, 25, and 50, respectively. Again, arbitrary and chosen based on our experience constructing this type of model.

We note that as the CBIABM is focused on properties of innate immunity and does not include expansion of the immune response to T-cell functions and other aspects of the adaptive response, it is incorrect to try and fit mediator trajectories to clinical time points much beyond 10–14 days (the time-period before cytotoxic T-cell populations peak and is associated with potential cytokine storm [31]). However, since the induction of T-cell responses is itself a function of the ability to either contain or propagate initial inflammatory signals and/or contain the degree of viral replication, we believe that the conclusions we can draw about the differential tolerance between bat and human immune properties are valid.

Following calibration, simulation experiments were carried out for both human parameterizations and bat parameterizations. The first set of experiments were a sweep of the initial viral inoculum from a value of 25 to 150 (arbitrary units) in increments of 25 and evaluating CBIABM behavior and outcome after 14 simulated days (*n* = 1000 stochastic replicates per initial inoculum value). These simulations were used to establish dose-dependent population outcome curves reflecting end-state %System-Health.

The calibration process nearly always uncovers insufficiencies in the model; this is an expected part of model development. Sometimes this can lead to insights that can guide further model development. An example of this occurred during the development of the CBIABM, where we noted that when the CBIABM only incorporated the differences between bat and human [2,3,4] within epithelial tissue and immune cells (specifically, incorporation of inflammasome functions in macrophages), there were not significant differences in bat versus human responses to viral infection in terms of final system health. Specifically, just increasing the “metabolic-byproduct” slightly beyond a two-fold increase in the bat parameterization resulted in similar final %System-Health levels compared to human parameterizations. This was substantially less difference between the two systems than expected, where in the real world, the > ten-fold increase in bat metabolic rate (represented by our “metabolic byproduct” variable) results in negligible effects from viral infection on bat health. Our analysis of these simulations suggested that the impaired inflammasome activation and increased antiviral effect of type 1 interferons in the bat parameterization of the CBIABM appeared to be offset by the enhanced eradication of infected epi-cells in the human parameterization of the CBIABM, resulting in a greater similarity of the total dead epi-cell agents generated than anticipated. This could be explained by the fact that the primary means of controlling viral infections is through the eradication of infected cells and that both the primarily apoptotic mechanisms in the bat and the inflammasome-affected processes in the human had similar efficacy in depleting the infected epithelial population with minimal excess “collateral damage” due to the increased proinflammatory processes in the human. It was only when the process of differential endothelial activation was added, which is related to inflammasome activation and for which we assumed would similarly be reduced in bats, that noticeable differences in bat resistance to viral load became evident. In order to characterize this effect, a parameter sweep was run across progressing thresholds of endothelial activation (from the human threshold = 5 to the bat threshold = 10; all arbitrary units) in the human parameterization of the CBIABM, using the highest tested viral initial inoculum (= 150) to visualize the greatest effect.

Finally, one of the proposed mechanisms for zoonotic transfer from bat populations to humans is viral spillover as a result of increased stress in bat populations (either due to environmental changes or the introduction of other types of infections, such as white-nose syndrome [5]). We simulated this effect by varying the parameter value of metabolic stress (“metabolic byproduct”) in the bat parameterization of the CBIABM and evaluated trends in population %System-Health (*n* = 1000 stochastic replicates per initial condition) associated with those changes.

## 3. Results

### 3.1. Initial Calibration

Initial calibration of the human parameterization of the CBIABM was able to generate the following plausible behaviors: viral dynamics that resulted in peaks at 3–7 days, decreases in %System-Health consistent with the onset of symptoms at ~3 days, and elevated mediators at 3–7 days consistent with the development of symptoms attributable to the presence of those mediators. Examples showing time course data for 10 stochastic replicates each at simulated moderate (initial inoculum = 75) and severe (initial inoculum = 150) infection are displayed in Figure 2, below and in Appendix D Figure A1. Figure 2 shows trajectories in terms of %System-Health (Panel A) and Extracellular Virus (Panel B). %System-Health is a proxy for disease severity and symptoms, with the onset of symptoms occurring around day 3. In Panel A %System-Health decreases thereafter due primarily to antiviral cell-killing until a nadir is reached at ~7 days before recovery starts. We note that since the simulation is never “killed”, the capacity to recover is always present as long as a single epithelial cell is still alive. However, for purposes of showing differential consequences to simulated viral infection, the inability of the CBIABM to “die” is not relevant. Figure 2B shows the additional calibration data that were generated with respect to mediator time course data. The same representative runs of *n* = 10 stochastic replicates for each moderate disease severity and severe disease severity provide the mediator trajectories shown in Appendix D. All these trajectories show plausible dynamics, rising with immune sensing of virus and infected epithelial cells and plateauing with/if containment of the infection. The values are unitless, which does not affect the qualitative shape of these trajectories, which is the relevant behavior being evaluated for plausibility. Note also that with the lack of T cells in the current version of the CBIABM, their absence results in the lack of their contribution in terms of generating inflammation-related mediators in the >7 day period of simulated time.

### 3.2. Parameter Sweep of Initial Viral Inoculum

The sweep of viral initial inoculum is seen in Figure 3A,B. The results from these and subsequent sweeps are presented in such a fashion to better visualize and compare the population-level outcomes for a particular set of initial conditions. Because of the stochasticity of the CBIABM (which mimics the heterogeneity seen in biological and clinical data sets), different individual runs will produce different results resulting in the population distribution of the outcome. In order to aid in the visual comparison of these population distributions, we plot the results of individual runs in ranked in terms of their end %System-Health on the *X*-axis (with each value on the *X*-axis representing an individual run) from the highest values of the output metric to the left and then by decreasing order moving rightwards along the *X*-axis: we term this a “ranked-order population distribution.” Figure 3A depicts the comparison between the human parameterization and the bat parameterization of the CBIABM. There is an expected dose-dependent relationship between the initial inoculum and the degree of reduced %System-Health. There is a considerably greater degree of impaired %System-Health seen in the human parameterizations such that bat parameterizations show minimal tissue damage (surrogate for the manifestation of disease and disease severity) for levels of initial inoculum that produce considerably reduced %System-Health in the human parameterizations. The resistance of the bat parameterizations is so marked that the dose-dependent differences to the varied initialinoculum can only be appreciated in the magnified view seen in Figure 3B.

### 3.3. Parameter Sweep of Endothelial Activation Threshold

The parameter sweep of the endothelial activation threshold (a proxy for the ability to activate the endothelial inflammasome) can be seen in Figure 4; these simulations were all carried out with an initial inoculum = 150 in order to depict the greatest impact of varying this parameter. Figure 4 demonstrates that progressively increasing the threshold for endothelial activation toward levels seen in the bat (= 10) in otherwise human-parameterized versions of the CBIABM shows increased resistance to viruses. However, while the human-parameterized version with bat-level endothelial activation threshold is substantially more resistant, it still does not match the resistance level in the bat parameterization. While this suggests that the other features of increased bat tolerance to viral infection do play a role in reducing the manifestation of disease in bats (e.g., differences in T1IFN properties), the considerable reduction in population disease severity seen only by reducing endothelial inflammasome activity points to it as a potentially impactful mechanism for bat viral tolerance.

### 3.4. Investigation into Effect of Metabolic Stress on Potential Viral Spillover

The last set of simulation experiments are intended to investigate the potential stress-related causes of viral spillover in bat populations, a presumed process involved in the zoonotic transfer that can lead to pandemics [5]. We translate the more general concept of “stress” into a representation in metabolic terms, and therefore represent increased stress level by increasing the metabolic byproduct (Met-By) variable in the CBIABM. We performed a parameter sweep of Met-By, starting from the value in the baseline bat parameterization (= 2.0, itself a 10× increase over the human parameterization) and increasing it by increments of 1; as with the parameter sweep for endothelial activation, all these simulations were run with an initial inoculum = 150. The results of this parameter sweep can be seen in Figure 5, with progressively worse %System-Health population distributions with increasing values of metabolic byproduct (e.g., increasing stress).

We believe that increased manifestation of disease/disease severity for equivalent degrees of viral exposure would lead to increased viral shedding and lead to viral spillover and increase the chance for the viral spillover that could lead to zoonotic transfer.

## 4. Discussion

The recent viral pandemics involving Ebola, SARS, MERS, and, most recently and dramatically, SARS-COV2/COVID-19 have placed focus on bats as a zoonotic reservoir for viruses that are candidates to drive the next pandemic. In particular, the unique physiological adaptations for flight seen in bats has led to interest in the intersection of the ability of bats to deal with the metabolic consequences of flight and their noted viral tolerance, with resulting insights into how those adaptations have manifested in differences in their innate immune response and dampened inflammatory response to cellular injury. This finding is of particular importance in understanding the potential pathophysiological manifestations should bat-borne viruses acquire the ability to infect human tissues, as there is compelling evidence that clinical disease severity in viral infections is heavily driven by the human host’s inflammatory response (e.g., “cytokine storm”) [6,32,33,34,35,36]. This has led to a great deal of interest in studying particular molecular components (the inflammasome [6]) and cellular functions (pyroptosis [23]) that can lead to increased inflammation and more severe disease. While most recent work has understandably focused on COVID-19/SARS-COV2, the role of inflammation-induced collateral tissue damage is a shared feature of many viral infections [6,32,33,34,35,36]. We pose that developing response capabilities to future pandemics should identify countermeasures focused on conserved mechanisms that drive viral disease severity. Such therapeutic countermeasures aimed at disease mitigation would fill a gap between public health non-pharmacological interventions (i.e., contact/transmission limiting strategies) and virus-specific modalities such as vaccines or new therapeutics that target virus-specific mechanisms of infectivity. Despite the impressive and unprecedented success and rapidity of COVID-19 vaccine development, it is difficult to imagine how such modalities could be made available in less than a year. More generic pandemic countermeasures would target shared mechanisms of viral pathophysiology, of which inflammation-propagating mechanisms represent a particularly attractive target for disease severity mitigation. Toward that end, we would pose that understanding how viral tolerance evolved and manifests in bats can produce insights into the design of targeted therapeutic immunomodulators to mitigate the severity of viral infection in humans.

We further contend that there is a potentially significant role for mathematical and computational modeling in the execution of comparative molecular biology. Currently, comparison from species to species is primarily descriptive; while there may be some intuitive impression about differences in molecular or cellular functions may manifest, there is a lack of formalized functional representations that allow for rigorous interrogation of highly complex dynamic systems. From one perspective, “comparative biology” is intrinsic to modern experimental biomedical research, in which animal models are used as surrogate systems to investigate human biology. However, the general goal here is not to emphasize the differences between animal models and humans. Instead, the focus is on similarities that allow findings from in vivo experimental systems to be extrapolated to human systems. Likewise, biomedical mathematical and computational models often rely on hypothesized similarities between laboratory/animal models and humans to inform their structure, suggest parameters, and offer validation. However, sometimes comparative biology instead focuses on characterizing the differences between an animal species and humans, most notably when a particular species might have a particular phenotype that could suggest a potential solution to a human disease process. One example is the resistance of elephants to cancer, despite their large size and long lifespan, which led to the finding of their increased DNA repair capacity due to their multiple copies of the p53 DNA repair/tumor suppression system [37,38]. The motivation of this current paper is another example of a zoonotically advantageous phenotype: the resistance of bats to viruses that cause significant disease in humans. Computational modeling projects that focus on these differences, especially those that seek to identify differences that might be targeted for therapeutic development, are much less common. While there has been interest in developing mathematical models of bat in-host viral dynamics (see [39] for a recent review noting specific papers modeling in-host bat viral dynamics [40,41,42,43]), to our knowledge, the CBIABM is the first computational model that explicitly represents the molecular processes that distinguish between bat and human responses to viral infections. Detailed models, such as the CBIABM, can explicitly represent both imputed differences between species and what is conserved from one context (be it a species or an individual) to the next. This capability also addresses a critical gap in the practice of biology, because as with the descriptive nature of differences between organisms/species, what is similar between organisms/species is often informally assumed (e.g., the justification for using wet-lab experimental models) but not rigorously and formally represented or evaluated [11].

The current version and associated experiments with the CBIABM represent how such a process might work. We make no claim that the CIABM is a comprehensive representation of either the bat or human immune system, nor even that it completely describes all the known features associated with bat viral tolerance. Rather, the CIABM is an explicit representation of a specific set of hypotheses related to a specific set of mechanisms proposed as being important in characterizing the difference between viral tolerance in bats and humans. As such, the CBIABM is not expected to be able to model a clinical disease course; it does not contain features, such as the adaptive immune system, that are known to play an important role in human disease, nor is it able to generate higher-order phenomena, such as vital signs or organ function, normally used to characterize disease. However, despite its abstraction, the CBIABM is a complex model (see Figure 1) and does represent a large list of known mechanisms identified through experimental biology. As such, the CBIABM can address questions about the role of those specific mechanisms it incorporates, a capability that is not present in more “theoretical” mathematical models that abstract away such detail so they may be more analytically tractable.

One example of an insight gained from this type of complex mechanism-based modeling is the potential role of the endothelial inflammasome: both as a feature that distinguishes between bat and human viral tolerance and as a potential therapeutic target in treating viral diseases. Most interest in the inflammasome has been understandably focused on its manifestation in immune cells [6,7,8,23,32,44]; this is reasonable since immune cells are the cellular effectors of the immune/inflammatory response. However, our simulations with the CBIABM show that in order to reflect the differences between bat and human responses to viral infection, it is insufficient to only focus on the immune cells. Comparison between bat and human parameterizations using only differences in immune cell inflammasome activity were able to generate some differences in viral tolerance, but not to the degree of the known differences between bat and human metabolic rates (>10-fold more in bats); see Figure 4. Alternatively, the endothelium is well recognized as playing a role in the acceleration of local tissue inflammation to systemic disease [27], but without a more “complicated” modeling context that parses the mechanism-based components of the immune response the impact of this feature on how bats maintain viral tolerance would not be appreciated. In our review of the literature, we were not able to find any studies examining bat endothelial inflammasomes but given the ubiquity of inflammatory cell adhesion and trafficking, we make the assumption that such molecular endothelial-mediated mechanisms do exist. We also reason that given the reduced inflammatory cell inflammasome activity seen in bats, such reduced activity would also be present in the inflammasome in other cell types. The results of these simulations with the CBIABM present a testable hypothesis that, if true, could guide a new approach to a single treatment applicable to a range of viral infections. In terms of potential therapeutic targets, while the endothelium in COVID-19 has received considerable attention, given that the receptor target for the SARS-COV2 spike protein is present in vascular tissue and given the clinically reported hypercoagulable complications (e.g., inappropriate clot formation) seen in COVID-19, many proposed non-vaccine therapies target the down-stream process of coagulation [45]. Alternatively, our comparative biology simulations suggest that a specific molecular mechanism/module, the endothelial inflammasome, could be a more general functional reason for why local viral infections become systemic and could potentially provide an area for investigating and developing more generalizable therapeutic agents. If future wet-lab experiments can be performed to support (or refute) the simulation findings from the CBIABM, then a significant advance in treating viral disease is possible.

We accept that there are limitations to the current version of the CBIABM. The CBIABM lacks representation of the adaptive immune response, which does not allow the CBIABM to replicate clinical disease dynamics and associated meditator time-series data much beyond 7–10 days post initial infection, at which time lymphocyte populations contribute heavily to the host response. This, in turn, limits the ability to use the CBIABM as a test platform for discovering new multi-modal and adaptive control strategies, which is one of the most potentially impactful roles for complex mechanism-based simulation models [46,47,48]. We further recognize that the current set of simulation experiments do not fully explore the different parameter combinations that might lead to the differences between viral tolerance phenotypes, with implications on the relative roles and contributions for cell-specific inflammasome activity and T1IFN antiviral functions.

In the future, we will directly address the limitations noted above. Components of the adaptive immune response will be added, first with attention to the cytotoxic acute phase components, then later incorporating those components and functions having to do with immune memory, both cellular and humoral. We will also incorporate more sophisticated representations of “stress” in order to examine the role of emergence from hibernation and the effect of co-infections on viral spillover. We will also perform a more comprehensive examination of model parameter space to gain greater insight into the combinatorial aspects that might generate different modes of viral tolerance. This will be done by adapting the CBIABM to our developed pipeline for using machine learning and artificial intelligence methods to refine model rules and parameter space while encompassing biological heterogeneity [30]. This approach, in concert with the modular structure of the CBIABM, will also allow future developments that include the implementation of virus-specific properties and functions related to invasiveness, replication and countermeasures to antiviral mechanisms, and the ability to represent differences in inflammasome suppression seen across various bat species [49].

## 5. Conclusions

We believe that there is considerable benefit in the comparative biology study of the differences between bats and humans with respect to viral tolerance and that these investigations can provide invaluable insights that may help our preparations for the inevitable “next pandemic.” We also believe that comparative computational models of sufficient complexity and representation of biological mechanisms, such as the CBIABM, can serve as important adjuncts to help inform and integrate more traditional experimental and investigatory methods and help develop biological countermeasures that could mitigate the health and societal impact of future pandemics.

## Figures and Tables

**Figure 1 viruses-13-01620-f001:**
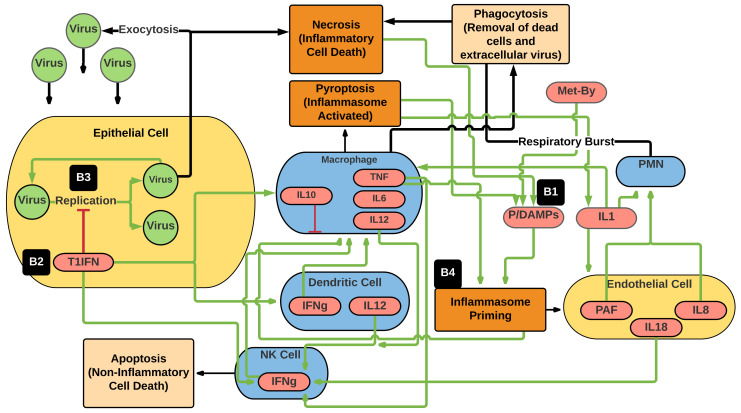
Schematic of the main cell types, mediators, interactions, and functions represented in the CBIABM. Blue ovals represent immune cell types; yellow ovals represent non-mobile cells; green circles represent virus; red ovals are molecular species; beige rectangles represent non/anti-inflammatory processes; orange rectangles represent proinflammatory processes. Green arrows represent positive or stimulatory interactions; red connectors represent negative or inhibitory interactions; black arrows represent cellular functions facilitated by the connected cell types. Note that the primary means of suppressing viral infection is through the death of infected epithelial cells, either by apoptosis or necrosis; both pathways lead to decreased numbers of healthy epithelial cells (%System-Health). In addition, note that the differences in bat and human parameterization are seen at: B1 = increased P/DAMPS by the addition of higher metabolic byproduct (Met-By) reflecting increased metabolism in bats, B2 = baseline production of type 1 interferons (T1IFN), B3 = enhanced antiviral effect of T1IFNs and B4 = decreased inflammasome activity. T1IFN = type 1 interferons, P/DAMPS = pathogen/damage-associated molecular patterns, Met-By = metabolic byproduct, PAF = platelet-activating factor, IL1 = interleukin 1, IL2 = interleukin 2, IL6 = interleukin 6, IL10 = interleukin 10, IL12 = interleukin 12, TNF = tumor necrosis factor, IFNg = interferon-gamma, PMN = polymorphonuclear neutrophil, NK cell = natural killer cell.

**Figure 2 viruses-13-01620-f002:**
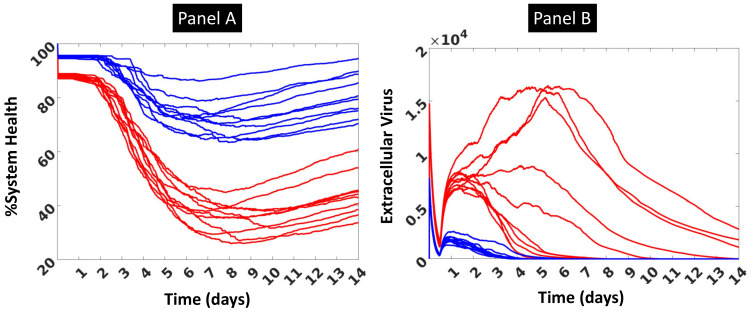
Representative trajectories of 14 days simulated time produced by the human parameterization of the CBIABM with 10 stochastic replicates at moderate disease severity (initial inoculum = 75 shown in blue lines) and severe disease severity (initial inoculum = 150 shown in red lines). Panel (**A**) shows %System-Health over time, with decrements starting just before day 3 and nadiring at ~day 7 before some recovery occurs. Panel (**B**) shows levels of extracellular virus over time. High initial values represent initial inoculum, of which only a percentage will lead to cellular invasion (see Epithelial Cell Rule 1a), which accounts for the rapid drop. Viral levels rise during incubation, which peak and then are suppressed in the moderate disease severity and sometimes not so in the severe disease severity. The values for extracellular virus on the Y-axis do not have units, as they are not relevant to depict the range of trajectories shown. These simulations demonstrate the effect of stochastic processes in variation in the trajectories, consistent with the heterogeneity present within biological data.

**Figure 3 viruses-13-01620-f003:**
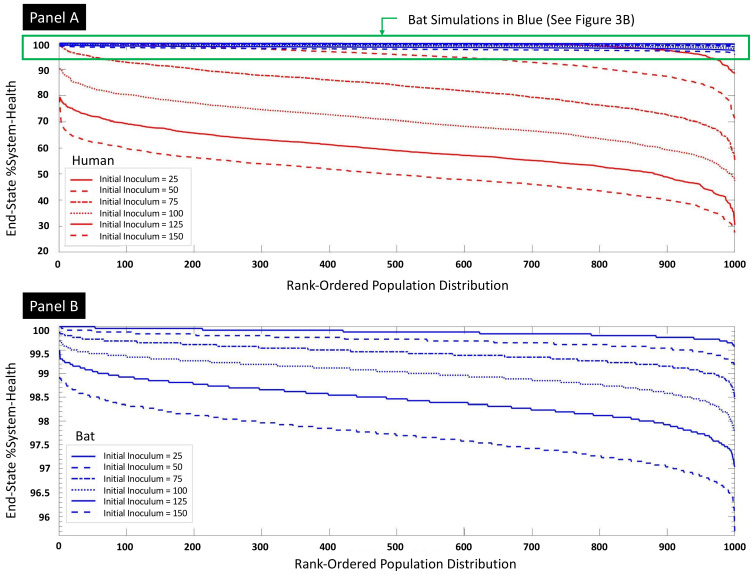
Sweeps of virus initial inoculum for bat and human parameterizations with 1000 stochastic replicates per condition. Panel (**A**) shows both bat (blue) and human (red) rank-ordered population distributions at initial inoculums from 25 to 150 in 25 increments. The rank-ordered population distributions reflect the stochastic replicates ranked by their end #System-Health at the end of 14 days simulated time; while these lines are technically 1000 individual columns, for visualization purposes, they are shown as curves. The decreasing curves seen with increasing initial inoculum are consistent with a dose-dependent worsening of outcomes across a population of runs. Note that the bat parameterizations essentially demonstrate no disease severity (in terms of reduced %System-Health) for the corresponding IIs that generate significantly reduced %System-Health in human parameterizations. Panel (**B**) is a magnification of the results for the bat parameterizations, which shows a similar dose-dependency but with considerably reduced tissue damage (note Y = axis for Panel B starts at 96).

**Figure 4 viruses-13-01620-f004:**
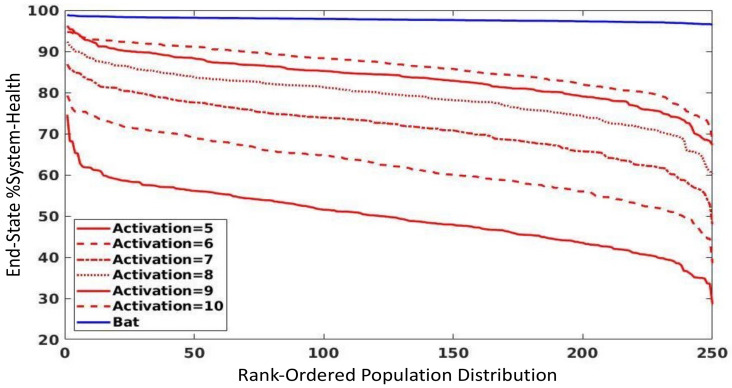
Parameter sweep of endothelial activation threshold level from baseline human parameterization (= 5) to baseline bat parameterization level (= 10) with initial inoculum = 150, 14 days of simulated time. The endothelial activation threshold reflects the ability to activate the endothelial inflammasome, with subsequent effects (adhesion activation and PAF production) that recruit circulating PMNs to the area of activation. There is a progressive reduction in the degree of tissue damage/disease severity across simulated populations (stochastic replicates *n* = 250) with increasing endothelium activation thresholds (simulating decreasing endothelial inflammasome activation), though even when the endothelium activation threshold is at the same level (= 10) between the bat- and human parameterizations, there is still an increased tolerance of the bat versions to viral insult.

**Figure 5 viruses-13-01620-f005:**
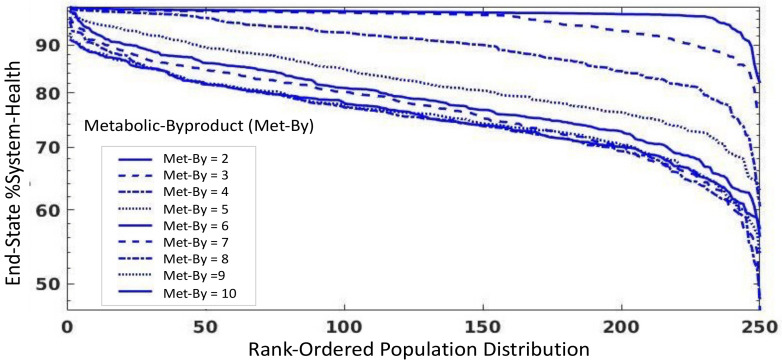
Parameter sweep of metabolic byproduct (Met-By), a proxy variable for stress in the CBIABM. All these simulations were carried out with an initial inoculum = 150 and run for 14 days of simulated time. The value for Met-By in the baseline bat parameterization is itself a 10× increase in the same term in the human parameterization, reflecting the increased metabolic stress from powered flight. Increasing Met-By demonstrates a progressive worsening of the population distribution of %System-Health, which we consider a pre-condition for increase viral shedding. Interestingly, the decreasing %System-Health seen with increasing Met-By appears to converge monotonically.

## Data Availability

Not applicable.

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
