# Peer review of "Comparative Computational Modeling of the Bat and Human Immune Response to Viral Infection with the Comparative Biology Immune Agent Based Model"

_viruses, 2021, doi:10.3390/v13081620_

Round 1

Reviewer 1 Report

See the attached docx file.

Author Response

Please see attachment (and note that responses to both reviewers are included in the document)

Reviewer 2 Report

Overall, I feel this is a well-written well motivated manuscript that introduces mathematically modelling in a novel area and in an interesting way. The Comparative Biology Immune Agent-Based Model (CBIABM) is proposed as a way of investigating differences between species (in this case humans and bats) and using the known biological differences between species to infer the driving forces behind disease trajectories of viral injections. 

While, the article is of interest to readers of Viruses, there is need for some revisions and clarity in places to make this work reliable, impactful and also reproduceable. The suggestions/comments are sectioned into either "Major" or "Minor" suggested revision: 

Major comments

  • Can more support from the literature be given both in the introduction and in the model development to support the use of computational modelling in hypothesis testing and the use of SEIR models? There is one citation on line 116 supporting the point around why computational modelling is advantageous and SEIR modelling and there are many very impactful SEIR models in the literature that should be cited here. 
  • The description of the Comparative Biology approach and "dynamic knowledge representation" is confusing and needs to be clearer. There are many examples in the literature of models that have been used for hypothesis testing, similar to what is done in this manuscript, that should be cited. In turn, if the authors feel there aren't other works in the literature that use a similar approach, then maybe they can contrast their approach to others in the literature to highlight how this is different for the reader, and this would also help the reader to understand their approach.
  • Can the authors provide more information on their fitting methodology, parameter values, references and data fitting plots? The authors say "the parameters and rules for the CBIABM are thus fitting to generate behaviours in realistic time courses" but I couldn't find the parameters that were fit or any of the model calibration? This information needs to be provided for readers and to also demonstrate how the model was validated. 
  • I think the description for the model rules is not sufficient. For example, natural killer cells are said to follow chemotaxis to T1IFN but there is no description for how chemotaxis is modelled and in general how cells move in the domain? This is the case in a lot of places where the assumption or description is given but no mathematical insight or parameter values or rates. Is this because these are pre-built into NetLogo? If so this needs to be better described and brief descriptions should still be given in places.  For example the susceptible to infection, is that a constant probability or does is changes as a function of extracellular intracellular density? How is this probability determined?
  • The authors should include plots for all variables in the model over time, for example the virus, the infected cell/dead cell numbers, the NKs, the macrophages, etc, otherwise it is not possible for the readers to know whether the dynamics of the immune cells match their understanding. In addition, I believe a plot of the Netlogo ABM dynamics over time should also be given to assist readers in understanding what the model looks like. 
  • A clearer description needs to be given for what the "Rank-Ordered Population Distributions" are. I read the description in the caption, but I still feel more in-text description is needed. For example, Fig 3, seems to suggest that for large inoculations in the human scenario that for 1000 simulations of the CBIABM that we can see anywhere from 80%-25% system health, which is quite a large variation in the outcome of the simulation, why does this occur? Is there any reliability in this much variability?
  • While I completely agree with the authors point around how this model is used as a comparitative biology testing tool and a way to investigate the biological differences between bats and humans, the results still feel lacking in validation with human/bat data. As such, I feel either this data/support needs to be provided (as it might be done when the fitting information is given) and/or a more extensive parameter investigation needs to be done. For example, how can we be sure that the assumptions in the human case and bat case differ this drastically and there shouldn't be a spectrum of dynamics between the two?

Minor comments:

  • I think this sentence may be missing a word, see line 149-151: “For the CBIABM, the intent is to evaluate the functional consequences of differing responsiveness several functional modules in the immune response to viral infection between humans and bats.”
  • What are the units of viral inoculum and how were these initial ranges determined? E.g. line 416. Is every variable in the model unit less? If so can the authors argue why this doesn't affect their insights and conclusions?
  • I think the sentence on lines 420-423 may need to be reworded, or there may be a typo, it is a bit confusing.
  • What are the units or the interpretation of “level” on lines 435-436
  • Could you label or title the figures in Figure 2 with their corresponding variables so it’s easier to determine immediately what is plotted instead of reading through the abstract. In turn can the authors add (a), (b), (c) etc to the image?
  • If the authors want to use "II" in the legend of Figure 2, they should put what it means in the caption and maybe in the text?
  • Should this is “of” instead of “if” in line 506?
  • The figures are very grainy, is there anyway their quality could be improved?

Author Response

Please see attachment, note that the document includes responses to both reviewers

Round 2

Reviewer 2 Report

Very happy with the improvements the authors made. I feel they have significantly improved the manuscript